# A multicentral prospective cohort trial of a pharmacist-led nutritional intervention on serum potassium levels in outpatients with chronic kidney disease: The MieYaku-Chronic Kidney Disease project

Yuki Asai[1,2]*, Asami Muramatsu[3], Tatsuya Kobayashi[4], Ikuhiro Takasaki[5], Toshiki Murasaka[6], Ai Izukawa[7], Kahori Miyada[8], Takahiro Okazaki[9], Tatsuki Yanagawa[2], Yasuharu Abe[10], Yasushi Takai[11], Takuya Iwamoto[1]

1 Department of Pharmacy, Faculty of Medicine, Mie University Hospital, Mie University, Tsu, Mie, Japan, 2 Pharmacy, National Hospital Organization Mie Chuo Medical Center, Tsu, Mie, Japan, 3 Nutrition Management Office, National Hospital Organization Mie Chuo Medical Center, Tsu, Mie, Japan, 4 Mie Pharmaceutical Association Kaiei Hisai Dispensing Pharmacy, Tsu, Mie, Japan, 5 Hisai Center Pharmacy, Tsu, Mie, Japan, 6 Konan Pharmacy, Tsu, Mie, Japan, 7 Ai Pharmacy Myojin Store, Tsu, Mie, Japan, 8 Sugi Pharmacy Hisai Intergarden Store, Tsu, Mie, Japan, 9 Department of Cardiology, National Hospital Organization Mie Chuo Medical Center, Tsu, Mie, Japan, 10 Mie Pharmaceutical Association, Tsu, Mie, Japan, 11 Department of Pharmacy, Mie Heart Center Hospital, Taki, Mie, Japan

* yuki0715asai@gmail.com

**Data Availability Statement:** The raw laboratory data supporting the findings of this study are

## Abstract

Although dietary potassium restriction is an acceptable approach to hyperkalemia prevention, it may be insufficient for outpatients with chronic kidney disease (CKD). Most outpatients with CKD use community pharmacies owing to the free access scheme in Japan. The MieYaku-CKD project included a community pharmacist-led nutritional intervention for dietary potassium restriction, with the goal of determining its efficacy for patients' awareness of potassium restriction and serum potassium levels in outpatients with CKD. This was a five-community pharmacy multicenter prospective cohort study with an open-label, before-and-after comparison design. Eligible patients (n = 25) with an estimated glomerular filtration rate (eGFR) < 45 mL/min/1.73 m$^2$ received nutritional guidance from community pharmacists. The primary outcome was a change in serum potassium levels at 12 weeks post-intervention. The eligible patients' knowledge, awareness, and implementation of potassium restriction were evaluated using a questionnaire. The median value of serum potassium was significantly reduced from 4.7 mEq/L before to 4.4 mEq/L after the intervention [*p* < 0.001, 95% confidence interval (CI): 0.156–0.500], with no changes in eGFR (*p* = 0.563, 95% CI: -2.427–2.555) and blood urine nitrogen/serum creatinine ratio (*p* = 0.904, 95% CI: -1.793–1.214). The value of serum potassium had a tendency of attenuation from 5.3 to 4.6 mEq/L (*p* = 0.046, 95% CI: 0.272–1.114) in the eGFR < 30 mL/min/1.73 m$^2$ group. A questionnaire revealed that after the intervention, knowledge and attitudes regarding dietary potassium restriction were much greater than before, suggesting that the decrease in serum potassium levels may be related to this nutritional guidance. Our findings indicate that

openly available in repository Mendeley Data at
https://data.mendeley.com/datasets/c9kydmj6c2/1.
DOI is 10.17632/c9kydmj6c2.1.

**Funding:** This work was supported by grant of
Japan Pharmaceutical Association "Yakuzaishi
shokuno shinko kenkyu josei jigyo"(grant for
research and studies which seek to develop
pharmacy profession and function in healthcare
and pharmaceutical affairs) (Grant No. jpa2022-
03). The funders had no role in study design, data
collection and analysis, decision to publish, or
preparation of the manuscript.

**Competing interests:** The authors have declared
that no competing interests exist.

implementing a dietary potassium restriction guidance program in community pharmacies is feasible and may result in lower serum potassium levels in outpatients with CKD.

## 1 Introduction

Hyperkalemia is a life-threatening metabolic problem that can lead to cardiac arrest in extreme cases [1]. Patients with chronic kidney disease (CKD) often acquire this condition, with an incidence rate of 14–20% in patients with CKD [2]. Serum potassium levels $\geq$ 5.5 mEq/L in CKD may be associated with increases in acute- [3] and long-term [4] mortality. Therefore, hyperkalemia is the most important adverse event that shortens lifetime in patients with CKD. The mechanism of developing hyperkalemia is attributed to be reduced kidney function because serum potassium is primarily extracted via urine. In fact, during CKD stage 3b [5], urinary excretion rate is substantially reduced, and the occurrence of hyperkalemia events is elevated [3]. While the prevalence of hyperkalemia in patients with CKD has recently increased due to increased prescription of renin-angiotensin system inhibitors (RASi) [6, 7], discontinuation of RASi [8], or mineralocorticoid receptor antagonists [9] due to hyperkalemia may be associated with a higher risk of death and major adverse cardiovascular events; therefore, even drugs that cause hyperkalemia should be continued for the long-term prognosis of patients with CKD.

Dietary potassium restriction is a reasonable approach for preventing hyperkalemia and has been advised for patients with CKD worldwide [10]. A meta-analysis revealed that dietary potassium restriction can lower serum potassium levels [11]. However, because of influencing factors that alter serum potassium levels, such as hydration level, acid-base status, glycemic management, and gastrointestinal complications, it is challenging to identify the appropriate potassium intake for specific patients with CKD [12]. Further complicating potassium restriction is the fact that different foods have different potassium contents [13], and severe potassium restriction in the diet (restricting vegetable and fruit intake) has been associated with death regardless of CKD stage [14]. Various lifestyle habits (cigarette smoking, infrequent alcohol consumption, and low physical activity) and characteristics (younger age, higher body mass index, male gender, and lower educational attainment) may be involved in low-adherence to CKD-specific dietary recommendations, including potassium restriction [15], indicating that guidance regarding potassium restriction needs to be individualized to fit the patient's lifestyle and characteristics. Therefore, it is crucial that dietary potassium restriction be managed by healthcare workers in conjunction with patients with CKD.

The role of the pharmacist extends beyond dispensing medications and plays an essential role in health promotion and screening [16]. There are 61,715 community pharmacies in United States, and 48.1% people live within 1 mile of any pharmacy [17]. Therefore, it goes without saying that community pharmacies are highly accessible healthcare locations for many outpatients with CKD. Hyperkalemia episodes occurred in 47.3% of the outpatients [3], indicating that community pharmacists can help outpatients with CKD in restricting their potassium intake. To the best of our knowledge, no information exists on the effects of community pharmacist-led nutritional interventions on serum potassium levels in outpatients with CKD, therefore, we hypothesized that the nutritional guidance provided by community pharmacists for outpatients with CKD may be a useful technique for dietary potassium restriction.

The present study aims to investigate the influence of community pharmacist-led nutritional guidance about dietary potassium restriction on the patient awareness and serum

potassium levels in outpatients with CKD in a community setting (MieYaku Chronic Kidney [MY-CKD] Project).

## 2 Martials and methods

### 2.1 Study design

This was a multicenter prospective cohort study with an open-label, before-and-after comparison design. This study was conducted in five community pharmacies belonging to the Tsu Pharmaceutical Association. Cardiologists, registered dieticians, and pharmacists from the National Hospital Organization Mie Chuo Medical Center (Mie, Japan) participated in this study.

### 2.2 Participants

Urinary potassium excretion decreases estimated glomerular filtration rate (eGFR) $< 45$ mL/min/1.73 m$^2$ in the Japanese population [5], therefore, the cut-off value for eligible patients with CKD was set at eGFR $< 45$ mL/min/1.73 m$^2$. The inclusion criterion was adult outpatients (age $\geq 18$ years old) at Mie Chuo Medical Center with eGFR $< 45$ mL/min/1.73 m$^2$. As a result, the following patients were excluded from the study: patients (a) below the age of 18, (b) with communication difficulties, (c) with hypokalemia (serum potassium level $< 4.0$ mEq/L), (d) undergoing hemodialysis, (e) taking potassium preparations, and (f) who received a refill prescription If the patient brought a refill prescription, serum potassium level could not be evaluated because there is no need to visit the Mie Chuo Medical center during or after the 12-week follow-up period. Therefore, "those who received a refill prescription" was set as an exclusion criterion.

### 2.3 Outcomes

All registered patients were followed up for 12 weeks. The following patients were excluded from the outcome evaluation during the observation period: those with (1) changes in the dosage of drugs affecting serum potassium levels, such as RASi, mineralocorticoid receptor antagonists, loop diuretics, or potassium-binding agents; (2) hemodialysis introduction; and (3) failure to complete the questionnaires at 12 weeks.

The primary outcome was the change in serum potassium levels from baseline (pre-intervention, day 1) to 12 weeks (post-intervention, day 84). eGFR and blood urine nitrogen (BUN)/serum creatinine ratio were assessed as indicators of renal function and hydration [18], respectively. The development of acute kidney injury (AKI) was elucidated during the 12-week follow-up period. An AKI was defined as an increase of at least 0.3 mg/dL in creatinine within 48 h or a 1.5-fold increase in creatinine, which is known or presumed to have occurred within 7 days [19]. Additionally, the difference between sodium and chloride levels (Na-Cl) mEq/L as a surrogate marker of metabolic acidosis in the present study [20]. In a subgroup study, the reduction of serum potassium levels was compared between eGFR $< 30$ and eGFR $\geq 30$ mL/min/1.73 m$^2$. Laboratory data were collected from Mie Chuo Medical Center's elective medical records.

The secondary outcome was the change in patients' attitudes toward potassium restriction at 12 weeks. Information about the questionnaire survey were listed in the S1 Methods. Questions 1 and 2 inquired about their understanding of hyperkalemia and potassium restrictions, respectively. Questions 3 and 4 were designed to address attitudes toward potassium restriction. Changes in the ratio between pre- and post-intervention were evaluated from questions 1

to 4. In the post-intervention period, questions 5 and 6 were meant to assess future awareness of potassium restriction.

## 2.4 Adverse event

The systolic and diastolic blood pressures (SBP and DBP) were monitored before and after the intervention. Cases of hypokalemia that required treatment were also collected.

## 2.5 Pharmacist intervention strategy

The intervention scheme is shown in Fig 1.

**2.5.1 Training.** On November 1, 2022, a cardiologist (TO), registered dietician (AM), and pharmacist (YA) gave a seminar to community pharmacists on potassium control in CKD. This educational program included a 60-min classroom session as well as a web-based component. Thirty-six community pharmacists attended the lectures and discussed their participation in the MY-CKD project. The purpose of this training was to standardize the quality of pharmacist interventions and eliminate study bias. Details of the lecture contents are provided in S1 Table.

**2.5.2 Intervention.** All the following interventions were implemented at each community pharmacy:

On the first intervention day (day 1), community pharmacists met with patients to describe the goals of the MY-CKD project, how to gather laboratory data, and lifestyle adjustments to avoid dialysis. Using a nutritional guidance sheet, community pharmacists lectured on dietary techniques for potassium restriction (S1 Fig). These nutritional guidance documents were

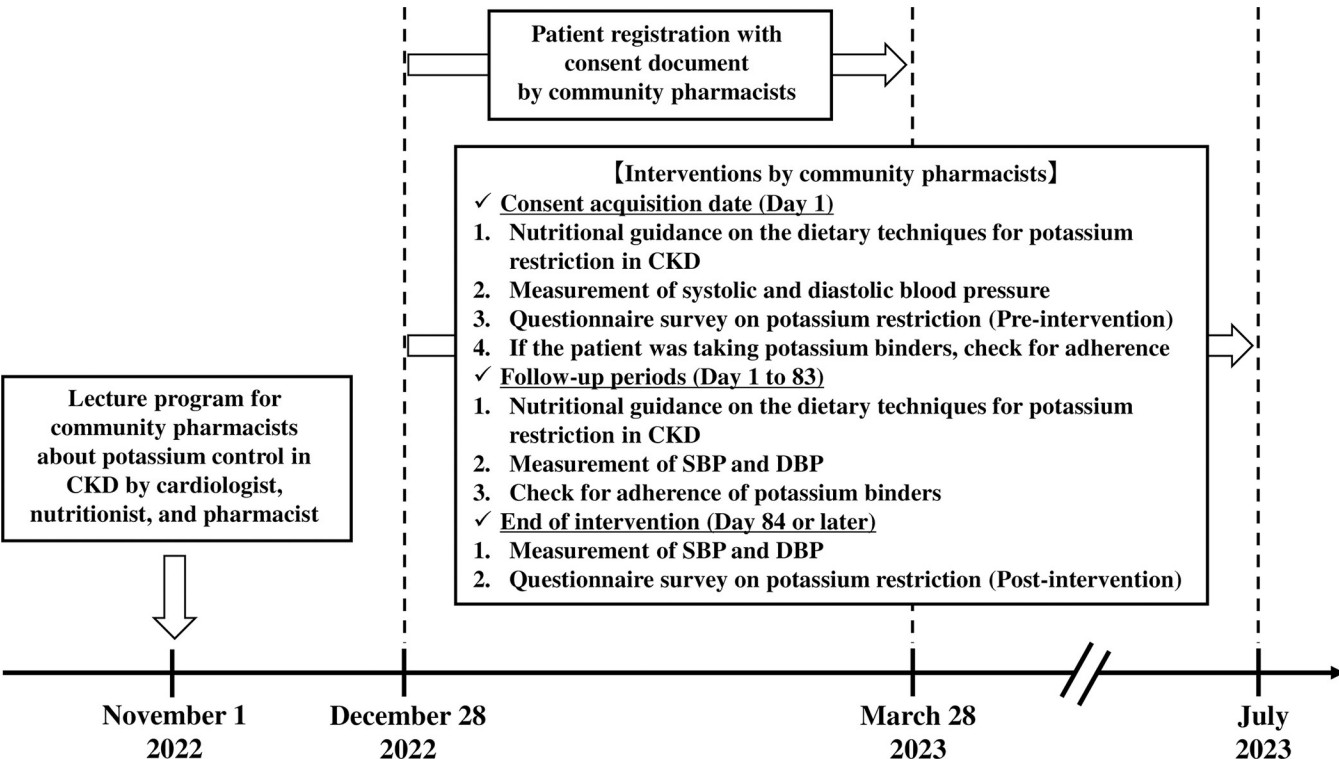

**Fig 1. Schematic illustrating the MieYaku-Chronic Kidney Disease (MY-CKD) project.** CKD, chronic kidney disease. DBP, diastolic blood pressure. SBP, systolic blood pressure.

mostly prepared by a registered dietician (AM) in consultation with healthcare workers. Following the lecture, a questionnaire was used to assess baseline awareness of potassium restriction, and SBP and DBP were measured at rest. Adherence was assessed if the patient was taking potassium-binding agents.

We continued to monitor the patient for dietary changes, kidney function, and serum potassium levels during the follow-up period.

A questionnaire survey to evaluate the awareness of potassium restriction was conducted 12 weeks after the intervention. SBP and DBP were measured at rest.

## 2.6 Sample size

In a prior study, nutritional advice for a potassium-restricted diet resulted in a decrease of up to 0.5 mEq/L in serum potassium levels [11]. The maximum standard deviation for serum potassium levels among individuals has been reported to be 0.8 mmol/L [21]. With a standard deviation of serum potassium levels of 1.0 mEq/L, and a mean difference of 0.5 mEq/L due to intervention, using a corresponding t-test ($\alpha = 0.05$ and $\beta = 0.20$), the number of cases necessary for the trial was calculated to be n = 34. Although similar reports have not existed, the estimated percentage dropouts over the observation period were set as 15.0% [22], the target number of registered patients was established as 40.

## 2.7 Statistical analysis

The Wilcoxon signed-rank test was used to examine differences in continuous variables, such as serum potassium, eGFR, BUN/creatinine ratio, Na-Cl, SBP, and DBP between pre- and post-intervention because respective variables followed a non-normal distribution. In the questionnaire survey, the rate of responses on the 4-point scale were compared pre- and post-intervention for questions 1 to 4, using the McNemar test. The effect size of changes in serum potassium level mediated by this intervention was calculated. All statistical analyses were performed using SPSS Statistics version 27 (IBM Japan, Tokyo, Japan), and the significance level was set at two-sided $p < 0.05$.

## 2.8 Ethics approval statement

This study was conducted in accordance with the Ethical Guidelines for Medical and Health Research Involving Human Subjects. The study protocol was approved by the National Hospital Organization, Mie Chuo Medical Center (approval ref. MCERB-202238) and Mie Pharmaceutical Association (approval ref. 2022–3). When community pharmacists referred to the value of eGFR on the prescription issued by Mie Chuo Medical Center, if they met the inclusion criteria, consent to participate in this study was obtained from the patients by each community pharmacist using a written consent document. All data were analyzed anonymously. We registered the protocol for this study in the University Hospital Medical Information Network Clinical Trials Registry (UMIN000049814).

## 3 Results

### 3.1 Characteristics of the participants

Enrolment in the MY-CKD project began on December 28, 2022, with five community pharmacies, and terminated on March 28, 2023. A total of 35 patients with CKD were resistant to treatment. Fig 2 shows that during the observation period, 25 patients were eligible for outcome evaluation based on the exclusion criteria, and the required number of patients was reached.

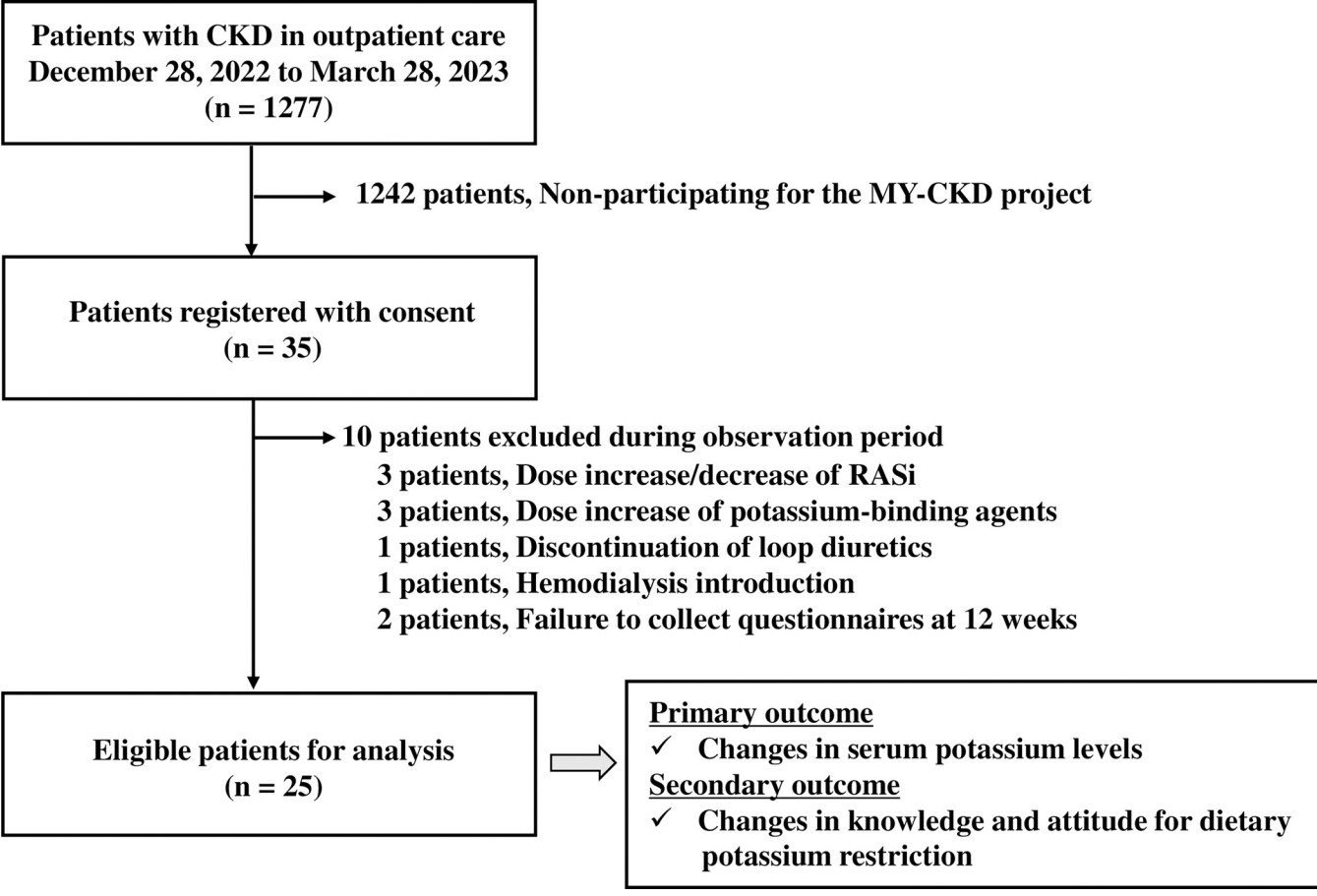

**Fig 2. Flowchart showing the selection of eligible patients.** CKD, chronic kidney disease. RASi, renin-angiotensin system inhibitors.

The baseline patient characteristics are presented in Table 1. There were 19 patients (64.0%) who were men, with a median age of 78 and a body mass index of 22.6 kg/m$^2$. The median eGFR was 35.0 mL/min/1.73 m$^2$ (range: 30.5–36.7 mL/min/1.73 m$^2$), which corresponds to the G3b stage. The baseline serum potassium level was 4.7 mEq/L (range: 4.3–5.1 mEq/L), which was close to the upper reference limit. While all patients cooked largely for themselves, only 8.0% (n = 2) requested advice from registered dietitians. Medication was self-managed by 80.0% of respondents. Only 12.0% (n = 3) of patients were taking potassium-binding agents.

### 3.2 Outcomes

Fig 3A shows that the median value of serum potassium significantly decreased from 4.7 mEq/L before intervention to 4.4 mEq/L post-intervention [$p < 0.001$, 95% confidence interval (CI): 0.156–0.500], and its effect size was -0.67. In contrast, eGFR ($p = 0.563$, 95%CI: -2.427–2.555) (Fig 3B) and BUN/creatinine ratio ($p = 0.904$, 95%CI: -1.793–1.214) (Fig 3C) did not change. In addition, the developed AKI was not observed during the 12-week follow-up period. There was no significant difference in the value of Na-Cl between pre- and post-intervention ($p = 0.377$, 95%CI: -1.324–0.444). In subgroup analysis, serum potassium levels had a tendency of attenuation in the eGFR < 30 mL/min/1.73 m$^2$ group (n = 6), from 5.3 mEq/L to 4.6 mEq/L ($p = 0.046$, 95%CI: 0.272–1.114) (Fig 4A). On the other hand, patients with

**Table 1. Baseline characteristics of eligible patients.**

| Factors | n = 25 |
|---|---|
| Basic property | |
| Male/Female, (Male, %) | 19/6 (76.0) |
| Age | 78 (70, 83) [a] |
| Body mass index (kg/m$^2$) | 22.6 (21.8, 24.9) [a] |
| Comorbidity | |
| Diabetes mellitus, n (%) | 17 (68.0) |
| Heart failure, n (%) | 13 (52.0) |
| Hypertension, n (%) | 22 (88.0) |
| Systolic blood pressure (mmHg) | 133 (125, 142) [a] |
| Diastolic blood pressure (mmHg) | 72 (65, 79) [a] |
| Laboratory data | |
| Na (mEq/L) | 139 (138, 140) [a] |
| Cl (mEq/L) | 106 (104, 107) [a] |
| Na-Cl (mEq/L) | 33 (32, 35) [a] |
| K (mEq/L) | 4.7 (4.3, 5.1) [a] |
| AST (IU/L) | 22 (21, 25) [a] |
| ALT (IU/L) | 16 (13, 22) [a] |
| Serum creatinine (mg/dL) | 1.5 (1.3, 1.7) [a] |
| eGFR (mL/min/1.73 m$^2$) | 35.0 (30.5, 36.7) [a] |
| BUN (mg/dL) | 27.7 (22.0, 30.4) [a] |
| BUN/creatinine ratio | 17.0 (13.6, 21.5) [a] |
| Hemoglobin (g/dL) | 12.9 (11.6, 13.6) [a] |
| Lifestyle | |
| Primary meal format at home | |
| Cooking by themselves, n (%) | 25 (100.0) |
| Required advice from registered dietitians, n (%) | 2 (8.0) |
| Medications | |
| Manage medicine | |
| Self-management, n (%) | 20 (80.0) |
| Family-management, n (%) | 5 (20.0) |
| RAS inhibitors, n (%) | 17 (68.0) |
| MRA, n (%) | 5 (20.0) |
| Loop diuretics, n (%) | 7 (28.0) |
| Potassium-binding agents, n (%) | 3 (12.0) |
| SGLT2 inhibitors, n (%) | 5 (20.0) |

ALT: alanine aminotransferase. AST: aspartate aminotransferase. BUN: blood urea nitrogen. eGFR: estimated glomerular filtration rate. MRA: mineralocorticoid receptor antagonist. RAS: renin-angiotensin system. SGLT2: sodium–glucose cotransporter 2. [a]Each value represents the median (25th, 75th percentile).

eGFR $\geq$ 30 mL/min/1.73 m$^2$ showed a significant decrease in serum potassium levels ($p = 0.009$, 95%CI: 0.057–0.364) (Fig 4B).

The percentage of responses changed significantly for all questions (Fig 5). For questions 1 and 2, no patients answered "Not at all" when asked about their understanding of the risk of hyperkalemia and potassium-rich foods ($p < 0.001$). Concerning routine potassium restriction awareness (question 3), the percentage of "Every time" increased ($p = 0.143$), while the percentage of "Not at all" decreased ($p = 0.008$). For question 4, "Is it bothersome to be aware of

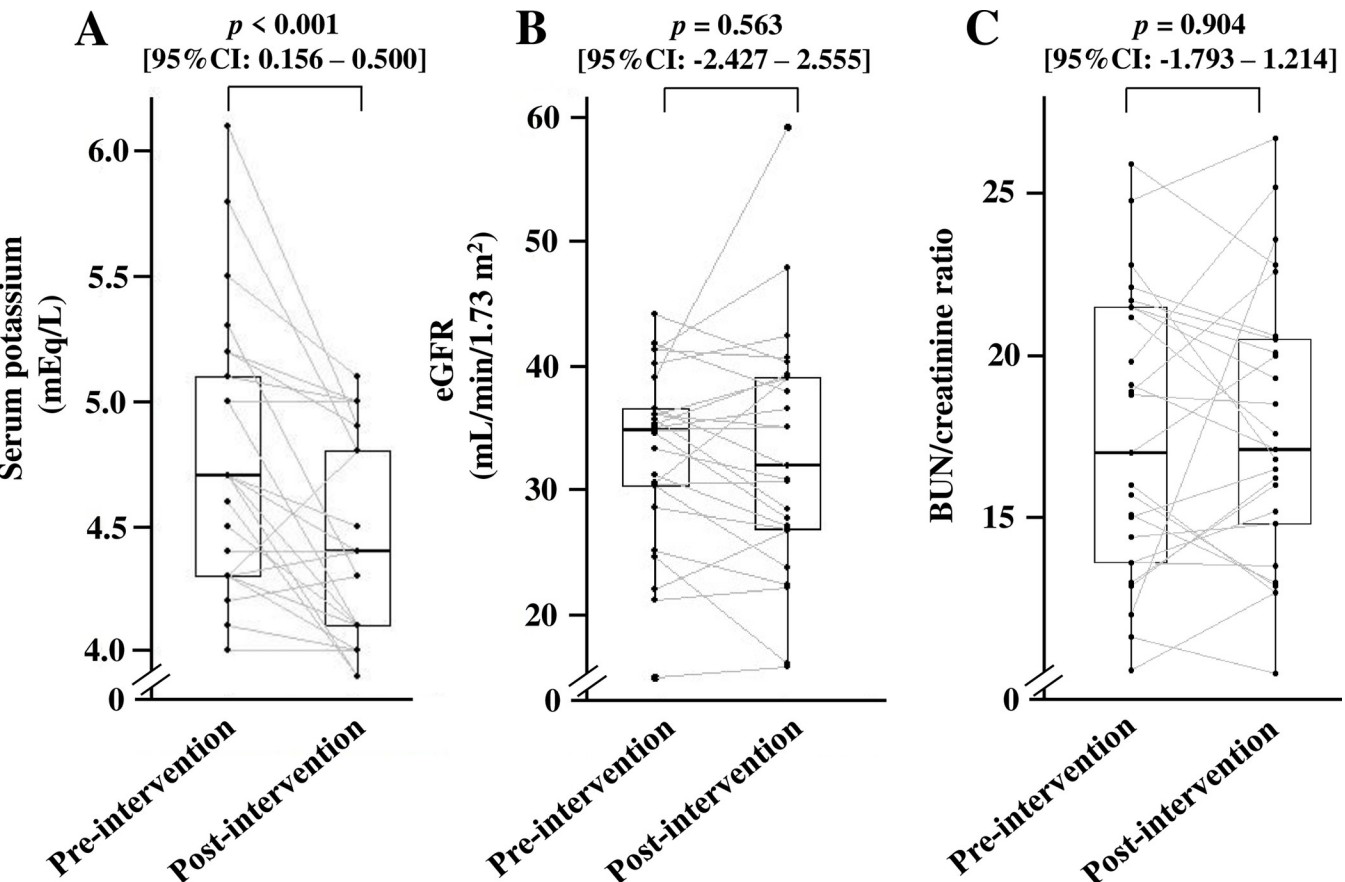

**Fig 3.** Effect of nutritional intervention on (A) serum potassium levels, (B) eGFR, and (C) BUN/creatinine ratio. The number of patients included in the analysis of each parameter was 25. BUN, blood urine nitrogen; eGFR, estimated glomerular filtration rate. The differences in respective variables between pre- and post-intervention were compared using the Wilcoxon signed-rank test.

your potassium intake?" the percentages who replied "Not at all" ($p < 0.001$) and "Not much" ($p = 0.012$) decreased, while the percentage who answered "Slightly" ($p = 0.136$) and "Extremely" increased ($p = 0.122$). "Frequently" was the most prevalent response to the question "Has this guidance made you more careful about your intake of foods containing potassium?", (56.0%, 14/25), followed by "Not that much," "Sometimes," and "Not at all." Finally, 68.0% (17/25) of respondents claimed they will continue to avoid consuming potassium-containing foods in the future.

### 3.3 Adverse event

Although the SBP and DBP of the six patients could not be followed in post-intervention, there were no significant changes in the median values of SBP or DBP following intervention (S2 Fig). Furthermore, there were no cases of hypokalemia requiring potassium supplementation.

### 4 Discussion

In the present study, serum potassium levels decreased without any changes in hydration levels, kidney function, or developing AKI. Furthermore, patients who underwent dose modification with concomitant drugs that affected serum potassium levels were excluded, suggesting

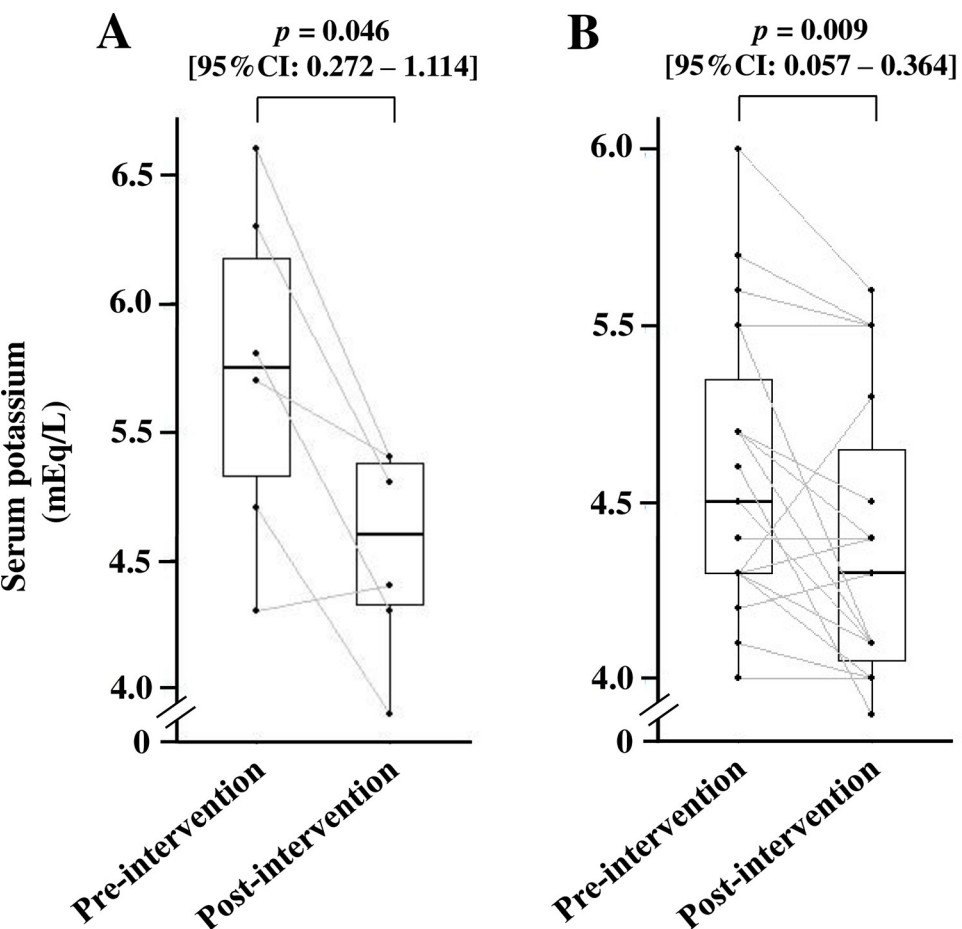

**Fig 4.** Changes in serum potassium levels in (A) eGFR < 30 mL/min/1.73 m$^2$ and (B) eGFR ≥ 30 mL/min/1.73 m$^2$. The number of patients in eGFR < 30 mL/min/1.73 m$^2$ and eGFR ≥ 30 mL/min/1.73 m$^2$ were six and 19, respectively. eGFR, estimated glomerular filtration rate. The differences in respective variables between pre- and post-intervention were compared using the Wilcoxon signed-rank test.

that the decrease in serum potassium levels could be attributed to nutritional guidance. The results of our questionnaire survey also support this hypothesis.

Although hyperkalemia is commonly defined as serum potassium levels ≥ 5.5 mEq/L, levels > 5.0 mEq/L have been linked to an increased risk of mortality in CKD stages 4–5 [23]. The reduction of serum potassium levels to less than 5.0 mEq/L in patients with eGFR < 30 mL/min/1.73 m$^2$ may help improve the prognosis of CKD patients; therefore, stricter control of serum potassium levels is required in the CKD stage of the patients in this study.

Patient knowledge of CKD is often minimal, with only 26.5% of patients with an eGFR < 60 mL/min/1.73 m$^2$ reporting awareness [24]. Moreover, even among hemodialysis patients, the level of knowledge acquisition regarding potassium restriction was poor [25]. Following the current nutritional intervention, 0.0% of patients answered "Not at all" when asked about the risk of hyperkalemia and foods with high potassium content (Fig 5), indicating increased knowledge among patients with CKD. Terlizzi et al. [26] found that multidisciplinary team support could help patients maintain a healthy nutritional status while delaying dialysis. The MY-CKD project's strengths were that community pharmacists learned the basics through lectures from cardiologists, registered dietitians, and hospital pharmacists (S1 Table),

**Q1. Do you know that high level of serum potassium can be dangerous?**

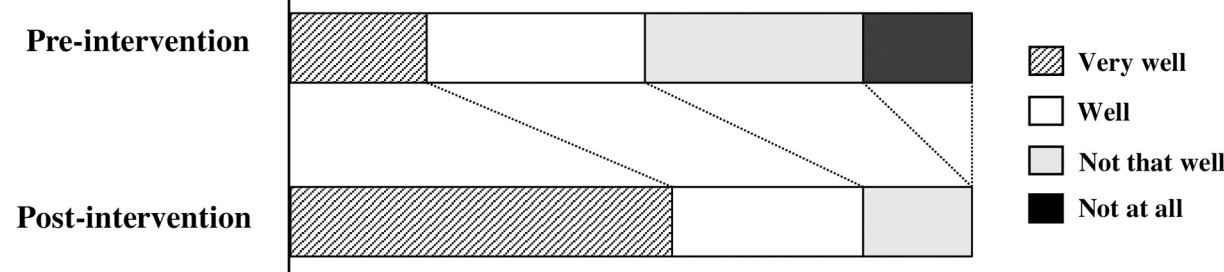

**Q2. Do you know what foods have high potassium content?**

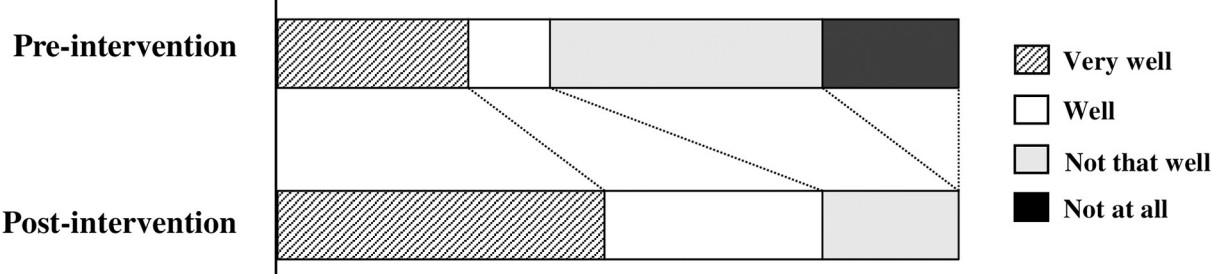

**Q3. Do you routinely regulate your consumption of foods with high potassium content?**

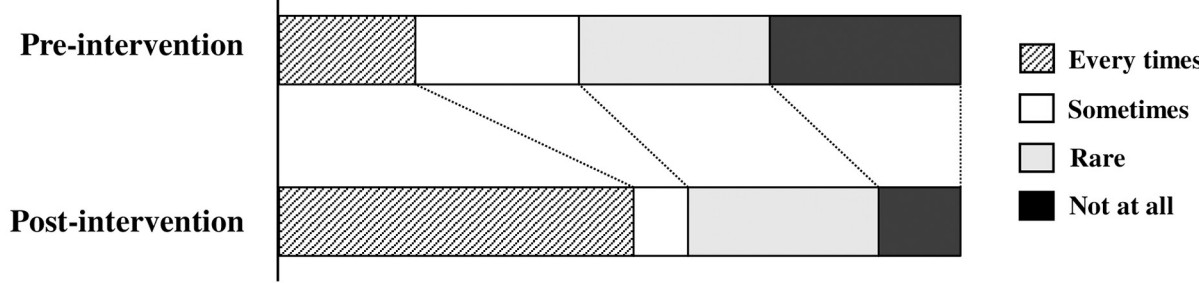

**Q4. Is it bothersome to be aware of your potassium intake?**

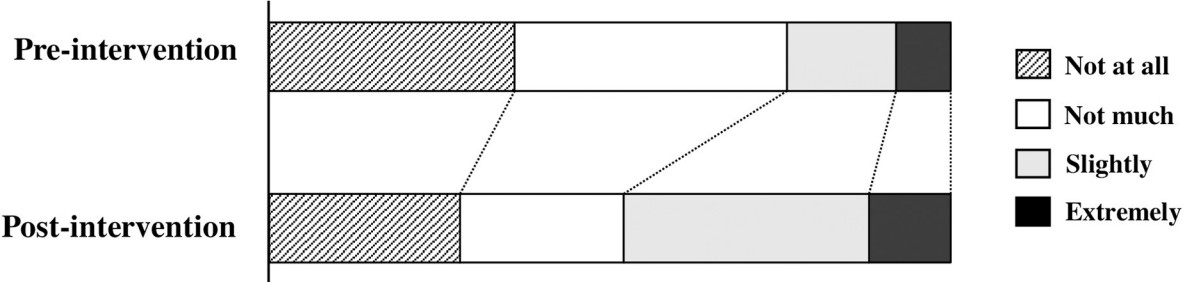

**Fig 5. Evaluation of changes in understanding and attitude regarding dietary potassium restriction obtained using a questionnaire survey.** The number of patients that responded to each question of the questionnaire was 25. The rate of responses on the 4-point scale were compared pre- and post-intervention for question 1 to 4, respectively, using the McNemar test.

and community pharmacists then passed on their knowledge to patients with CKD. Additionally, community pharmacists shared the goal of lowering serum potassium levels in patients with CKD, and constant follow-up on cooking procedures for potassium reduction was deemed important during the observation period. As all patients enrolled in this trial cooked for themselves (Table 1), this intervention strategy may result in a change in attitude toward routine potassium restriction. However, the percentage of patients with problematic potassium restriction increased (Fig 5). The MY-CKD project is thought to have provided the patient with information about the pathogenesis of CKD and potassium restriction. However, this may lead to patients needing nutritional management.

It is well-known that potassium intake is responsible for the dose-dependent reduction of blood pressure [27]. However, severe potassium restriction may cause blood pressure fluctuation in patients with CKD [28]. Therefore, the effect of dietary potassium restriction on SBP and DBP was classified as a negative event in this study. Neither SBP nor DBP changed significantly after the intervention (S2 Fig), suggesting that the degree of potassium restriction had no effect on blood pressure.

Our study had several limitations. First, the long-term persistence of the dietary potassium restriction could not be determined. Second, because the sample size was limited, and required sample size could not be reached, there could be unexplained confounding factors, such as patient selection bias. Third, in patients with eGFR $< 60$ mL/min/1.73 m$^2$ in Japan, the frequency of hyperkalemia was considerably higher in the winter (December–February) than that in the summer (June–August) [29]. Seasonal changes may have had an impact on 52.0% (13/25) of patients whose observation end date was in summer, indicating that seasonal variations cannot be ruled out. Fourth, the efficiency of this intervention strategy in patients undergoing hemodialysis remains unknown. Fifth, although the quality of the intervention method may be elevated with the participation of nephrologists, the National Hospital Organization Mie Chuo Medical Center did not have staff nephrologists. Therefore, it was considered necessary to construct a project with the participation of nephrologists in future investigations. Sixth, because the questionnaire used in the present study was developed originally, the relationship between changes in awareness/knowledge and specific dietary changes through questionnaires was not validated. Therefore, the extent to which the patients' diets have changed by the present questionnaire would not be fully evaluated. Finally, because this study only included patients with CKD who were outpatients at a secondary hospital in Japan, the impact on patients who attended hospitals with different characteristics, such as tertiary hospitals or clinics, is unknown. A cluster-randomized controlled study in Japan found that a lifestyle advising program in community pharmacies can lower SBP [30] and hemoglobin A1c [31]. Given these limitations and evidence, a cluster-randomized controlled design with long-term follow-up of potassium restriction is needed to evaluate the variability in serum potassium levels when patients are supported by community pharmacists.

CKD is one of the most important global public health problems that shortens the lifetime by increasing the requirement for dialysis treatment and risk of cardiovascular events [32]. The number of community pharmacies in Japan is comparable to those in the United States [17, 33], therefore, the evidence from the MY-CKD project of dietary potassium restriction by community pharmacist will provide great benefit for patients with CKD. Therefore, the present intervention strategy may serve as a model case in the world for nutritional guidance provided to CKD patients by community pharmacists.

## 5 Conclusion

This is the first study to show that a community pharmacist-led nutritional intervention mediated through the MY-CKD project improves dietary potassium restriction attitudes and lowers

serum potassium levels in patients with CKD. Although few studies have demonstrated the benefits of community pharmacist activities in practice, the implementation of nutritional interventions, such as potassium restriction, for patients with CKD by educated community pharmacists certainly contributes directly to the improvement of community populations' health.

## Supporting information

**S1 Checklist. CONSORT checklist.**
(DOC)

**S2 Checklist. STROBE statement—checklist of items that should be included in reports of *cohort studies.***
(DOCX)

**S1 File. Study protocol in English.**
(DOCX)

**S2 File. Study protocol in Japanese.**
(DOCX)

**S1 Methods. The questionnaire contents in MY-CKD project.**
(DOCX)

**S1 Table. Lecture contents in the MY-CKD project.**
(DOCX)

**S1 Fig. The nutritional guidance document of MY-CKD project.**
(PPTX)

**S2 Fig.** Changes in (A) SBP and (B) DBP by dietary potassium restriction. The number of patients in pre-intervention and in post-intervention were 25 and 19, respectively. DBP, diastolic blood pressure. SBP, systolic blood pressure. CI, confidence interval. The differences in respective variables between pre- and post-intervention were compared using the Wilcoxon signed-rank test.
(PPTX)

## Acknowledgments

We thank Tomoharu Hasebe, Hiroya Inui, Chie Suezawa, Zyun Niato, Akira Sato, Yuji Nakagawa, Yuko Niimi, Yasuki Ogino, Hiroki Sugino, Ryota Kobayashi, Yoshihiro Kinoshita, Takahiro Fukuyama, Koji Terada, Shinichi Maegawa, Kosuke Miyachi, Tomohiko Aoki, Taisuke Matsumuro, and Nobuyuki Nakagawa for their institutional enrollment in the MY-CKD project.

## Author Contributions

**Conceptualization:** Yuki Asai, Asami Muramatsu, Toshiki Murasaka, Takahiro Okazaki, Tatsuki Yanagawa, Yasuharu Abe, Yasushi Takai, Takuya Iwamoto.

**Data curation:** Yuki Asai, Tatsuya Kobayashi, Ikuhiro Takasaki, Toshiki Murasaka, Ai Izukawa, Kahori Miyada, Tatsuki Yanagawa.

**Formal analysis:** Yuki Asai, Takuya Iwamoto.

**Funding acquisition:** Yuki Asai.

**Investigation:** Yuki Asai, Tatsuya Kobayashi, Ikuhiro Takasaki, Toshiki Murasaka, Ai Izukawa, Kahori Miyada, Takahiro Okazaki, Tatsuki Yanagawa, Yasuharu Abe.

**Methodology:** Yuki Asai, Asami Muramatsu, Takahiro Okazaki, Tatsuki Yanagawa, Yasuharu Abe, Yasushi Takai, Takuya Iwamoto.

**Project administration:** Yuki Asai, Asami Muramatsu, Tatsuya Kobayashi, Ikuhiro Takasaki, Toshiki Murasaka, Ai Izukawa, Kahori Miyada, Takahiro Okazaki, Tatsuki Yanagawa, Yasuharu Abe, Yasushi Takai.

**Resources:** Yuki Asai.

**Writing – original draft:** Yuki Asai.

**Writing – review & editing:** Yasuharu Abe, Yasushi Takai, Takuya Iwamoto.

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
