## [Decision Letter · Decision Letter 0]

2 Feb 2024

PONE-D-23-33112A multicentral prospective cohort trial of a pharmacist-led nutritional intervention on serum potassium levels in outpatients with chronic kidney disease: The MieYaku-Chronic Kidney Disease projectPLOS ONE

Dear Dr. Asai,

Thank you for submitting your manuscript to PLOS ONE. After careful consideration, we feel that it has merit but does not fully meet PLOS ONE’s publication criteria as it currently stands. Therefore, we invite you to submit a revised version of the manuscript that addresses the points raised during the review process.

It is a very interesting study. Working on hyperkalemia by diet education is so important in patients with CKD.

I will ask you to review the article, and these are my comments:

Introduction:

1- First paragraph: the hyperkalemia in CKD can be elaborated in better way

2- Second paragraph: Diet restrictions in CKD: discus more the challenges faced

Study design:

1- Why the nephrologist didn’t participate to this study: Cardiologists, registered dieticians, and pharmacists from the National Hospital Organization Mie-Chuo Medical Center participated in this study.

2- What does this mean: those who received a refill prescription, in the exclusion criteria?

3- the patients who developed acute kidney injury during the 12 weeks of follow up, should also be excluded from the study or at least mentioned during the follow up, independently of the creatinine at 0 then 12 weeks

Outcomes:

1- Blood urine nitrogen levels should not be used as indicator of hydration

2- How they validate the questionnaire used. Validation is an important aspect in relation to the use of questionnaire/survey form.

3- Potassium level is affected by acidosis: this should also be studied/mentioned

The statistical analysis:

Correction of the t-test in the sample size

The estimated percentage dropouts (attrition rates) are to be stated.

Wilcoxon signed-rank test and McNemar test are not clear. More information is to be provided.

One or two-sided p value is to be mentioned.

Review the figures.

We look forward to receiving your revised manuscript.

Kind regards,

Chadia Beaini, M.D.

Guest Editor

PLOS ONE

Journal Requirements:

2. Please provide a completed TREND checklist (http://www.cdc.gov/trendstatement/)

3. Thank you for stating the following financial disclosure:"This work was supported by grant of Japan Pharmaceutical Association "Yakuzaishi shokuno shinko kenkyu josei jigyo"(grant for research and studies which seek to develop pharmacy profession and function in healthcare and pharmaceutical affairs) (Grant No. jpa2022-03)."

Reviewers' comments:

Reviewer's Responses to Questions

**Comments to the Author**

1. Is the manuscript technically sound, and do the data support the conclusions?

Reviewer #1: Partly

2. Has the statistical analysis been performed appropriately and rigorously? 

Reviewer #1: No

3. Have the authors made all data underlying the findings in their manuscript fully available?

Reviewer #1: Yes

4. Is the manuscript presented in an intelligible fashion and written in standard English?

Reviewer #1: Yes

5. Review Comments to the Author

Reviewer #1: Line 98, to assess the secondary outcome ‘change in patients’ attitudes toward potassium restriction at 12 weeks’ using 2 questions (Q3 & Q4) and understanding of hyperkalemia and potassium restrictions respectively (Q1 & Q2) is insufficient. To measure these outcomes usually requires a series of questions.

Line 127, information on whether the questionnaire/questions was/were validated or piloted is to be mentioned. Validation is an important aspect in relation to the use of questionnaire/survey form.

Line 139, alpha and beta figures are incorrectly cited. It should be alpha = 0.05, beta=0.20 (or power= 80%)

Line 140, the estimated percentage dropouts (attrition rates) are to be stated.

Line 144, the sentence is incomplete and requires revision.

Line 145, the use of McNemar is not clear. More information is to be provided.

Line 147, one or two-sided p value is to be mentioned.

For percentage figures in the text and tables, at least one decimal point is to be provided.

Figure 3, 4, 5, S2 Fig, the statistical test is to be denoted in the figure footnote. n to be stated.

Line 110, the sentence ‘This is the Fig 1 legend.' is to be omitted. Likewise with the sentence in Line 176, Line 178, Line 207, Line, 211, Line 213.

Effect size index and confidence interval could be reported.

Figure 1, the word ‘Questionary survey’ to be revised.

6. PLOS authors have the option to publish the peer review history of their article (what does this mean?). If published, this will include your full peer review and any attached files.

Reviewer #1: No

---

## [Author Response · Author response to Decision Letter 0]

14 Feb 2024

Response to Reviewers

We thank the reviewer for evaluating our manuscript and providing valuable suggestions. We have revised the manuscript in response to the comments.

Guest Editor: 

Introduction:

【Comment1】

1- First paragraph: the hyperkalemia in CKD can be elaborated in better way

【Answer1】

According to your suggestion, we have added the following sentences about hyperkalemia in CKD.

Lines 27-41

Hyperkalemia is a life-threatening metabolic problem that can lead to cardiac arrest in extreme cases [1]. Patients with chronic kidney disease (CKD) often acquire this condition, with an incidence rate of 14¬–20% in patients with CKD [2]. Serum potassium levels ≥ 5.5 mEq/L in CKD may be associated with increases in acute- [3] and long-term [4] mortality. Therefore, hyperkalemia is the most important adverse event that shortens lifetime in patients with CKD. The mechanism of developing hyperkalemia is attributed to be reduced kidney function because serum potassium is primarily extracted via urine. In fact, during CKD stage 3b [5], urinary excretion rate is substantially reduced, and the occurrence of hyperkalemia events is elevated [3]. While the prevalence of hyperkalemia in patients with CKD has recently increased due to increased prescription of renin-angiotensin system inhibitors (RASi) [6,7], discontinuation of RASi [8] or mineralocorticoid receptor antagonists [9] due to hyperkalemia may be associated with a higher risk of death and major adverse cardiovascular events; therefore, even drugs that cause hyperkalemia should be continued for the long-term prognosis of patients with CKD.

Reference list

2. Gilligan S, Raphael KL. Hyperkalemia and Hypokalemia in CKD: Prevalence, Risk Factors, and Clinical Outcomes. Adv Chronic Kidney Dis. 2017; 24: 315-318. 

4. Iseki K, Iseki C, Ikemiya Y, Fukiyama K. Risk of developing end-stage renal disease in a cohort of mass screening. Kidney Int. 1996; 49: 800-805. 

【Comment2】

2- Second paragraph: Diet restrictions in CKD: discus more the challenges faced

【Answer2】

We have inserted the following sentences.

Lines 42-58

Dietary potassium restriction is a reasonable approach for preventing hyperkalemia and has been advised for patients with CKD worldwide [10]. A meta-analysis revealed that dietary potassium restriction can lower serum potassium levels [11]. However, because of influencing factors that alter serum potassium levels, such as hydration level, acid-base status, glycemic management, and gastrointestinal complications, it is challenging to identify the appropriate potassium intake for specific patients with CKD [12]. Further complicating potassium restriction is the fact that different foods have different potassium content [13], and severe potassium restriction in the diet (restricting vegetable and fruit intake) has been associated with death regardless of CKD stage [14]. Various lifestyle habits (cigarette smoking, infrequent alcohol consumption, and low physical activity) and characteristics (younger age, higher body mass index, male gender, and lower educational attainment) may be involved in low-adherence to CKD-specific dietary recommendations, including potassium restriction [15], indicating that guidance regarding potassium restriction needs to be individualized to fit the patient's lifestyle and characteristics. Therefore, it is crucial that dietary potassium restriction be managed by healthcare workers in conjunction with patients with CKD.

Reference list

15. Kaesler N, Baid-Agrawal S, Grams S, Nadal J, Schmid M, Schneider MP, Eckardt KU, Floege J, Bergmann MM, Schlieper G, Saritas T. Low adherence to CKD-specific dietary recommendations associates with impaired kidney function, dyslipidemia, and inflammation. Eur J Clin Nutr. 2021; 75: 1389-1397. 

Study design:

【Comment3】

1- Why the nephrologist didn’t participate to this study: Cardiologists, registered dieticians, and pharmacists from the National Hospital Organization Mie-Chuo Medical Center participated in this study.

【Answer3】

As you mentioned, the quality of the intervention method may be elevated with the participation of nephrologists. However, the National Hospital Organization Mie Chuo Medical Center did not have a staff nephrologist. Therefore, it was considered necessary to construct a project with the participation of nephrologists in the future investigation. 

According to your suggestion, we have inserted the following sentences.

Lines 304-308

Fifth, although the quality of the intervention method may be elevated with the participation of nephrologists, the National Hospital Organization Mie Chuo Medical Center did not have staff nephrologists. Therefore, it was considered necessary to construct a project with the participation of nephrologists in future investigations.

【Comment4】

2- What does this mean: those who received a refill prescription, in the exclusion criteria?

【Answer4】

If the patient brought a refill prescription, serum potassium level cannot be evaluated because there is no need to visit the Mie Chuo Medical center during or after the 12-week follow-up period and. Therefore, “those who received a refill prescription” was set as an exclusion criterion.

According to your indication, we have added the following sentences.

Lines 91-95

If the patient brought a refill prescription, serum potassium level could not be evaluated because there is no need to visit the Mie Chuo Medical center during or after the 12-week follow-up period. Therefore, “those who received a refill prescription” was set as an exclusion criterion.

【Comment5】

3- the patients who developed acute kidney injury during the 12 weeks of follow up, should also be excluded from the study or at least mentioned during the follow up, independently of the creatinine at 0 then 12 weeks

【Answer5】

We conducted the additional survey to clarify the incidence of acute kidney injury (AKI) during the 12 weeks of follow up. An AKI was defined as an increase in creatinine of at least 0.3 mg/dL within 48 hours or a 1.5-fold increase in creatinine, which is known or presumed to have occurred within 7 days [KDIGO Clinical Practice Guideline for Acute Kidney Injury].

No patient developed AKI during the 12-week follow-up period, suggesting that AKI would not affect the serum potassium levels during 12-week follow-up periods.

According to your suggestion, we have included the following sentences and the baseline of serum creatinine (Table 1).

Materials and Methods section

Lines 106-109

The development of acute kidney injury (AKI) was elucidated during the 12-week follow-up period. An AKI was defined as an increase of at least 0.3 mg/dL in creatinine within 48 h or a 1.5-fold increase in creatinine, which is known or presumed to have occurred within 7 days [19].

Result section

Lines 216-217

In addition, the developed AKI was not observed during the 12-week follow-up period.

Discussion section

Lines 260-261

In the present study, serum potassium levels decreased without any changes in hydration levels, kidney function, or developing AKI.

Reference list

19. Kidney Disease Improving Global Outcomes: KDIGO Clinical Practice Guideline for Acute Kidney Injury. Section 2: AKI Definition. 2012. [Cited 2024 Feb 10]. Available from: https://kdigo.org/wp-content/uploads/2016/10/KDIGO-2012-AKI-Guideline-English.pdf.

Outcomes:

【Comment6】

1- Blood urine nitrogen levels should not be used as indicator of hydration

【Answer6】

Blood urine nitrogen (BUN) and serum creatinine ratio have been used as indicators of hydration [18]. Therefore, we evaluated the hydration status by changes in BUN/creatinine ratio. The significance differences were not observed between pre- and post-intervention (p = 0.904, 95% confidence interval (CI): -1.793 – 1.214).

According to your suggestion, we have revised the following sentences and Figure 3C. In addition, the baseline of BUN/creatinine ratio was added in Table 1.

Abstract section

Lines 16-17

blood urine nitrogen/serum creatinine ratio (p = 0.38, 95% CI: -1.793–1.214)

Materials and methods section

Lines 104-106

eGFR and blood urine nitrogen (BUN)/serum creatinine ratio were assessed as indicators of renal function and hydration [18], respectively.

Result section

Lines 214-216

In contrast, eGFR (p = 0.563, 95%CI: -2.427 – 2.555) (Fig 3B) and BUN/creatinine ratio (p = 0.904, 95%CI: -1.793–1.214) (Fig 3C) did not change.

Lines 235-239

Fig. 3. Effect of nutritional intervention on (A) serum potassium levels, (B) eGFR, and (C) BUN/creatinine ratio. The number of patients included in the analysis of each parameter was 25. BUN, blood urine nitrogen. eGFR, estimated glomerular filtration rate. CI, confidence interval. The differences in respective variables between pre- and post-intervention were compared by the Wilcoxon signed-rank test. 

Reference list

18.　Robinson BE, Weber H. Dehydration despite drinking: beyond the BUN/Creatinine ratio. J Am Med Dir Assoc. 2004; 5: S67–S71. 

【Comment7】

2- How they validate the questionnaire used. Validation is an important aspect in relation to the use of questionnaire/survey form.

【Answer7】

The questionnaire used in the present study was not validated because we originally developed it. The relationship between changes in awareness/knowledge and specific dietary changes through questionnaires should be evaluated, but has not been done. Therefore, the extent to which the patients' diets have changed by the present form of questionnaire would not be fully evaluated. 

According to your suggestion, we have added the following sentence in limitation.

Lines 308-312

Sixth, because the questionnaire used in the present study was developed originally, the relationship between changes in awareness/knowledge and specific dietary changes through questionnaires was not validated. Therefore, the extent to which the patients' diets have changed by the present questionnaire would not be fully evaluated.

【Comment8】

3- Potassium level is affected by acidosis: this should also be studied/mentioned.

【Answer8】

In this study, blood gas test was not available because the study was conducted on outpatients. Alternately, the difference between sodium and chloride levels (Na-Cl) mEq/L as a surrogate marker of metabolic acidosis in the present study [20]. There was no significant difference in the value of Na-Cl between pre- and post-intervention, therefore changes in status of acidosis might not be associated with the potassium level alteration.

According to your suggestion, we have added the following sentences. In addition, the baseline value of Na-Cl was added in Table 1.

Materials and Methods section

Lines 109-111

Additionally, the difference between sodium and chloride levels (Na–Cl) mEq/L as a surrogate marker of metabolic acidosis in the present study [20].

Result section

Lines 217-218

There was no significant difference in the value of Na-Cl between pre- and post-intervention (p = 0.377, 95%CI: -1.324–0.444).

The statistical analysis:

【Comment9】

Correction of the t-test in the sample size

【Answer9】

There was a typographical error in the sample size calculation from study protocol. Thank you for your understanding.

It has been corrected as follows.

Sample size section

Lines 158-163

With a standard deviation of serum potassium levels of 1.0 mEq/L, and a mean difference of 0.5 mEq/L due to intervention, using a corresponding t-test (α = 0.05 and β = 0.20), the number of cases necessary for the trial was calculated to be n = 34. Although similar reports have not existed, the estimated percentage dropouts over the observation period were set as 15.0% [22], the target number of registered patients was established as 40.

Moreover, the following sentences has been revised.

Lines 297-299

Second, because the sample size was limited, and required sample size could not be reached, there could be unexplained confounding factors, such as patient selection bias.

【Comment10】

The estimated percentage dropouts (attrition rates) are to be stated.

【Answer10】

We have revised the following sentences.

Lines 161-163

Although similar reports have not existed, the estimated percentage dropouts over the observation period were set as 15.0% [22], the target number of registered patients was established at 40.

Reference list

22. Sedgwick P. Randomised controlled trials: the importance of sample size. BMJ. 2015; 350: h1586. doi: 10.1136/bmj.h1586.

【Comment11】

Wilcoxon signed-rank test and McNemar test are not clear. More information is to be provided.

【Answer11】

According to your suggestion, we have revised the following sentences.

Lines 166-170

The Wilcoxon signed-rank test was used to examine differences in continuous variables, such as serum potassium, eGFR, BUN/creatinine ratio, Na-Cl, SBP, and DBP between pre- and post-intervention because respective variables followed a non-normal distribution. In the questionnaire survey, the rate of responses on the 4-point scale were compared pre- and post-intervention for questions 1 to 4, using the McNemar test.

【Comment12】

One or two-sided p value is to be mentioned.

【Answer12】

As you indicated, we have revised the following sentence.

Lines 172-173

All statistical analyses were performed using SPSS Statistics version 27 (IBM Japan, Tokyo, Japan), and the significance level was set at two-sided p < 0.05.

 

Response to Reviewers

We thank the reviewer for evaluating our manuscript and providing valuable suggestions. We have revised the manuscript in response to the comments.

Reviewer #1:

【Comment1】

Line 98, to assess the secondary outcome ‘change in patients’ attitudes toward potassium restriction at 12 weeks’ using 2 questions (Q3 & Q4) and understanding of hyperkalemia and potassium restrictions respectively (Q1 & Q2) is insufficient. To measure these outcomes usually requires a series of questions.

【Comment2】

Line 127, information on whether the questionnaire/questions was/were validated or piloted is to be mentioned. Validation is an important aspect in relation to the use of questionnaire/survey form.

【Answer 1 and 2】

As you indicated, these were the only questions that could not adequately elucidate the changes in patient knowledge/awareness. The questionnaire used in the present study was developed originally, therefore, the relationship between changes in awareness/knowledge and specific dietary changes through questionnaires was not validated. Therefore, the extent to which the patients' diets have changed by the present form of questionnaire would not be fully evaluated.

According to your suggestion, we have added the following sentence in limitation.

Lines 308-312

Sixth, because the questionnaire used in the present study was developed originally, the relationship between changes in awareness/knowledge and specific dietary changes through questionnaires was not validated. Therefore, the extent to which the patients' diets have changed by the present questionnaire would not be fully evaluated.

【Comment3】

Line 139, alpha and beta figures are incorrectly cited. It should be alpha = 0.05, beta=0.20 (or power= 80%)

【Answer3】

As you indicated, we have corrected the transcription error. 

Lines 158-161

With a standard deviation of serum potassium levels of 1.0 mEq/L, and a mean difference of 0.5 mEq/L due to intervention, using a corresponding t-test (α = 0.05 and β = 0.20), the number of cases necessary for the trial was calculated to be n = 34.

【Comment4】

Line 140, the estimated percentage dropouts (attrition rates) are to be stated.

【Answer4】

As you indicated, we have inserted the following sentences.

Lines 161-163

Although similar reports have not existed, the estimated percentage dropouts over the observation period were set as 15% [22], the target number of registered patients was established at 40.

Reference list

22. Sedgwick P. Randomised controlled trials: the importance of sample size. BMJ. 2015; 350: h1586.

【Comment5】

Line 144, the sentence is incomplete and requires revision. 

【Comment6】

Line 145, the use of McNemar is not clear. More information is to be provided.

【Answer 5 and 6】

According to your suggestion, we have revised the following sentences.

Lines 166-170

The Wilcoxon signed-rank test was used to examine differences in continuous variables

---

## [Decision Letter · Decision Letter 1]

14 May 2024

A multicentral prospective cohort trial of a pharmacist-led nutritional intervention on serum potassium levels in outpatients with chronic kidney disease: The MieYaku-Chronic Kidney Disease project

PONE-D-23-33112R1

Dear Dr. Asai,

We’re pleased to inform you that your manuscript has been judged scientifically suitable for publication and will be formally accepted for publication once it meets all outstanding technical requirements.

Kind regards,

Amir Hossein Behnoush

Academic Editor

PLOS ONE

Additional Editor Comments (optional):

Reviewers' comments:

Reviewer's Responses to Questions

**Comments to the Author**

1. If the authors have adequately addressed your comments raised in a previous round of review and you feel that this manuscript is now acceptable for publication, you may indicate that here to bypass the “Comments to the Author” section, enter your conflict of interest statement in the “Confidential to Editor” section, and submit your "Accept" recommendation.

Reviewer #1: All comments have been addressed

Reviewer #2: (No Response)

2. Is the manuscript technically sound, and do the data support the conclusions?

Reviewer #1: Partly

Reviewer #2: Yes

3. Has the statistical analysis been performed appropriately and rigorously? 

Reviewer #1: Yes

Reviewer #2: Yes

4. Have the authors made all data underlying the findings in their manuscript fully available?

Reviewer #1: Yes

Reviewer #2: Yes

5. Is the manuscript presented in an intelligible fashion and written in standard English?

Reviewer #1: Yes

Reviewer #2: Yes

6. Review Comments to the Author

Reviewer #1: (No Response)

Reviewer #2: The study titled "A multicentral prospective cohort trial of a pharmacist-led nutritional intervention on serum potassium levels in outpatients with chronic kidney disease: The MieYaku-Chronic Kidney Disease project" is well-written with accurate methodology and presentation of the results. I reviewed previous comments and the response of the authors to the comments. I have no other comments for improvement. Congratulations!

7. PLOS authors have the option to publish the peer review history of their article (what does this mean?). If published, this will include your full peer review and any attached files.

Reviewer #1: No

Reviewer #2: No

---

## [Editor Report · Acceptance letter]

22 May 2024

PONE-D-23-33112R1 

PLOS ONE

Dear Dr. Asai, 

I'm pleased to inform you that your manuscript has been deemed suitable for publication in PLOS ONE. Congratulations! Your manuscript is now being handed over to our production team.

Kind regards, 

on behalf of

Dr. Amir Hossein Behnoush 

Academic Editor

PLOS ONE